# Pedestrian Safety in Frontal Tram Collision, Part 1: Historical Overview and Experimental-Data-Based Biomechanical Study of Head Clashing in Frontal and Side Impacts

**DOI:** 10.3390/s23218819

**Published:** 2023-10-30

**Authors:** Frantisek Lopot, Lubos Tomsovsky, Frantisek Marsik, Jan Masek, Petr Kubovy, Roman Jezdik, Monika Sorfova, Barbora Hajkova, Dita Hylmarova, Martin Havlicek, Ondrej Stocek, Martin Doubek, Tommi Tikkanen, Martin Svoboda, Karel Jelen

**Affiliations:** 1Department of Anatomy and Biomechanics, Charles University, 162 52 Prague, Czech Republicmarsik@it.cas.cz (F.M.);; 2Department of Designing and Machine Elements, Czech Technical University, 166 29 Prague, Czech Republic; 3Institute of Thermomechanics, CAS, 182 00 Prague, Czech Republic; 4VUKV a.s., 158 00 Prague, Czech Republic; masek@vukv.cz (J.M.);; 5Dopravní Podnik Hlavního Města Prahy, 190 22 Prague, Czech Republic; doubekm@dpp.cz; 6GIM Oy, 02650 Espoo, Finland; 7Faculty of Mechanical Engineering, Jan Evangelista Purkyne University, Pasteurova 3544/1, 400 96 Usti nad Labem, Czech Republic; martin.svoboda@ujep.cz; 8Second Faculty of Medicine (2. LF UK), Charles University, 150 06 Prague, Czech Republic

**Keywords:** tram, crash test, front face design, windshield, pedestrian, safety, collision, head injury criterion

## Abstract

This article represents the first paper in a two-part series dealing with safety during tram–pedestrian collisions. This research is dedicated to the safety of trams for pedestrians during collisions and is motivated by the increased number of lethal cases. The first part of this paper includes an overview of tram face development from the earliest designs to the current ones in use and, at the same time, provides a synopsis and explanation of the technical context, including a link to current and forthcoming legislation. The historical design development can be characterised by three steps, from an almost vertical front face, to leaned and pointed shapes, to the current inclined low-edged windshield without a protruding coupler. However, since most major manufacturers now export their products worldwide and customisation is only of a technically insignificant nature, our conclusions are generalisable (supported by the example of Berlin). The most advantageous shape of the tram’s front, minimising the effects on pedestrians in all collision phases, has evolved rather spontaneously and was unprompted, and it is now being built into the European Commission regulations. The goal of the second part of this paper is to conduct a series of tram–pedestrian collisions with a focus on the frontal and side impacts using a crash test dummy (anthropomorphic test device—ATD). Four tram types approaching the collision at four different impact speeds (5 km/h, 10 km/h, 15 km/h, and 20 km/h) were used. The primary outcome variable was the resultant head acceleration. The risk and severity of possible head injuries were assessed using the head injury criterion (*HIC*_15_) and its linkage to the injury level on the Abbreviated Injury Scale (AIS). The results showed increasing head impacts with an increasing speed for all tram types and collision scenarios. Higher values of head acceleration were reached during the frontal impact (17–124 g) compared to the side one (2–84 g). The *HIC*_15_ values did not exceed the value of 300 for any experimental setting, and the probability of AIS4+ injuries did not exceed 10%. The outcomes of tram–pedestrian collisions can be influenced by the ATD’s position and orientation, the impact speed and front-end design of trams, and the site of initial contact.

## 1. Introduction

Recent years have seen economic development, population growth, and urbanisation all over the world [1,2]. Consequently, transport demands are expected to grow unabated, and according to the International Transport Forum (ITF), the total transport activity can more than double by 2050 compared to 2015 [3]. The number of individually owned motor vehicles reflects these increased transport demands, even despite the adverse health effects associated with said transportation [2]. Moreover, besides those impacts on individuals, cities contribute to three quarters of global energy consumption and greenhouse emissions [1,4]. Therefore, human-powered transport (walking and cycling) and public transport could benefit both the individual and the whole community and offer an interesting version of a sustainable future in urban agglomerations [1,2,3,4,5,6,7,8].

Primarily to mitigate the impacts of climate change and improve the environmental quality in cities (reducing air pollution, emissions, and traffic noise) while satisfying the needs of growing urban populations, the volume of public transport has been growing and actively promoted [1,2,4]. Rolling stocks have been found to be ecological, effective, and economical solutions to mobility in cities. Therefore, they could lead to improvements in urban sustainability and liveability [5]. However, the growing volume of public transport and the general accelerated pace and pressure of modern life coupled with pedestrians’ lack of attention (mainly due to the use of personal electronic devices) have resulted in an increased number of traffic accidents. Consequently, the issue of pedestrian safety is paramount [6,7,8]. The Prague Municipal Transport Company (DPP) keeps statistics on accidents where basic data for each event are registered. These data enable each accident to be associated with the tram type, accident location, traffic density, and a number of other factors.

Due to the reasons above, the development of tram fronts with regard to pedestrian safety in collisions and the campaign for uniform EU legislation to address this issue through a set of standards gradually started in the second half of the twentieth century. Globally, more than 400,000 people die each year in the US and EU [9]. While an increase in pedestrian deaths by 62–63% was observed between 1990 and 2010 [10], a slight decrease was already noted in some EU countries due to designs and the introduction of safety measures between 2010 and 2018 [11]. An analysis of accidents in the EU shows that the death rate of unprotected road users decreases gradually compared to the death rate of travellers in cars [12]. A comparison of accidents between trams and pedestrians with those between buses and pedestrians shows that the risk of fatal accidents is up to four times higher per 1 road km. In the total number of accidents, the risk of death is even 9 to 15 times higher [13]. Important factors that increase the risk of accidents overall include an inadequate traffic layout (parked cars around tram tracks) and the use of headphones by pedestrians while walking on streets in conjunction with the very low sound emission of trams in the direction of travel [14,15,16]). The statistics also show that a pedestrian is most often (50% of all cases) hit by the right third of the tram’s front closer to the right edge of the lane in the direction of travel (in countries with left-hand driving) [13,17,18,19,20]. Currently, the development of trams is no longer the exclusive issue of manufacturers. Rather, it has become an open issue in which a number of other research and developmental institutions and departments participate (e.g., the research and development project, Safe Tram Face of Škoda Transportation and West Bohemian University in Pilsen, from the years 2017 to 2021). All of the effort is focused on finding a concept for both the near and distant possibly autonomous future of this part of public transport.

Despite the considerable effort and financial investment made in the development and improvement of active traffic safety measures, such as collision avoidance systems, as well as passive safety measures (e.g., trams’ front design and construction), the number of fatalities and severe injuries has remained high [6,7,21,22]. Several studies have been conducted to analyse tram–pedestrian traffic accidents; however, the different methodological approaches and cities used in those studies hinder a sensible comparison of their results [22,23,24,25]. The study by Demant et al. [10] analysed tram–pedestrian accidents using the method of multidetector computed tomography (MDCT) for injury diagnosis in Cologne, Germany. Between September 2004 and December 2006, there were 18 people who were injured in tram accidents, which were retrospectively assessed. Two recurring injury patterns were identified: unilateral (*n* = 9) and complex (*n* = 9). In both injury pattern groups, head injuries were the most frequent (*n* = 15; 83%), followed by trauma-associated injuries in the thorax region (*n* = 12; 67%) and abdominal injuries (*n* = 8; 44%). Injuries in both groups also resulted in a total of eight amputations of extremities (*n* = 8; 44%). Four individuals died after the initial diagnosis (*n* = 4; 22%). Unfortunately, the study did not include the type of trams involved in each accident. The study by Szmaglinski et al. [23], conducted between 2013 and 2017 in the city of Gdansk, Poland, did not provide a total number of tram–pedestrian accidents or the type of trams involved in the collisions and accidents. However, collisions with pedestrians were found to be the second most common accident (15%). Although the study by Sagberg et al. [24], conducted between 1986 and 1996, also did not describe a tram type involved in tram–pedestrian accidents, 85% of accidents were those with pedestrians and 15% were those with bicyclists. The total collision incidence averaged 27.1 per year. There were ten fatalities, corresponding to about 7% of all tram–pedestrian accidents. The results also found that the pedestrians’ lack of attention towards the approaching tram was the most common cause of an accident (67%), which matched the outcomes of other studies [22,23]. The other common cause of an accident was a pedestrian crossing the street too close to the tram for the tram driver to notice it. A statistical analysis of tram–pedestrian accidents conducted in Prague, Czech Republic, between 2007 and 2020, found a total of 1233 accidents (Figure 1), out of which 56 were fatalities (5% on average; Figure 2) [25]. 

Similar to the previous studies, the analysis did not describe the tram type involved in each accident. However, the analysis showed a sharp increase in accidents during the years of 2017 and 2018. One possible explanation among many is that during these years, the number of old types of trams (T3R.PLF and KT8D5) was reduced, and they were replaced by modern types (14T and 15T) with a different design and construction of the tram’s front. The results also showed that although a total number of fatalities differed every year with a sharp increase in 2018, the relative number of fatalities indicated a decreasing trend compared to previous years [25]. The outcomes from 2019 and 2020 can be attributed to the COVID-19 pandemic. For this reason, an interpretation and comparison with previous years and other studies is limited and could lead to misleading conclusions. However, a post-COVID-19 time (2022) showed an increase in tram–pedestrian accidents and fatalities again. In addition, since 2016 (11), there have never been more than 10 severe injuries per year (i.e., a life-threatening injury or an injury requiring a long-term hospitalisation), but suddenly, there were 19 such injuries in 2022. Both of these findings confirmed a positive effect of the modern tram front-end design on lifting the pedestrian up in the collision and showed a potentially negative effect of too-rigid windshields.

Although trends in fatalities in tram–pedestrian traffic accidents exhibit a decreasing tendency, the frequency and total number of such accidents remain high. Therefore, the European Committee for Standardization (CEN) and rolling stock manufactures have started developing a methodology and technical report for a vehicle front-end design for trams and light rail vehicles that affect pedestrian safety [26]. The report is mainly focused on passive safety measures to reduce the consequences of tram–pedestrian collisions. The highest attention is paid to design recommendations for the front end of a vehicle to minimise the impact on a pedestrian when struck and to minimise the risk of being drawn under the vehicle. It also includes design recommendations for the frame under a vehicle to reduce severe injuries to the pedestrian lying on the tram tracks and recommendations to prevent a pedestrian from being run over by the wheels of the vehicle [26]. These recommendations only apply to new vehicles and consider just the side impact on a pedestrian. The report is also only focused on the primary (the initial contact of a pedestrian with the front end of a tram) and tertiary impacts (the risk of being run over by the wheels of a vehicle). It describes the geometric criteria to assess the severity of injuries and provides advice on how to perform a numerical simulation of a tram–pedestrian collision. However, the report does not define any criteria or methodologies for crash tests using ATDs. It only briefly describes the desired kinematics; it should favour either blocking the shoulder and the torso as quickly as possible while limiting the rotation of the torso or impacting a pedestrian progressively from the lower legs up to the torso and shoulders [26].

With this in mind, the aim of this study is to present a comprehensive overview of the development of tram-front designs and to provide a real-world experimental analysis of a tram–pedestrian collision using crash tests with an ATD in case of a frontal and side impact. It should thus enable the identification of the importance of individual parts and areas of the tram front design and construction. The outcomes of this study should also contribute to a better understanding of tram–pedestrian collisions and their reconstructions and improve technical reporting and methodologies for studying such traffic accidents. The authors hope that this work will consequently assist in the future design and development of new passive and active safety measures and technologies to increase the protection of pedestrians in urban areas. 

## 2. Development in Tram Front Face Design in Terms of Pedestrian Safety

Electric rail has been a feature of metropolitan public transport ever since the end of the 19th century. The development of human society in conjunction with expanded technical possibilities accelerated the growth of traffic density and passenger demands for speed, reliability, and comfort. Trams quickly evolved from the early open or semi-open designs to closed vehicles with an enclosed cabin for the driver. As trams became faster, it became crucial to address the issue of transport safety, primarily the safety of travellers. Trams therefore started to be equipped with closable entrances that only opened at stops. Increasing traffic density put pressure on the driver’s view field. The tram driver has to cover a significantly wider area compared to a car driver. The increasing population of cities has further led to a higher number of operating trams and the maximisation/optimisation of their interiors’ use (to accommodate as much travellers as possible). An increase in the number of trams operating concurrently required the search for a comprehensive solution to traffic management with respect to immediate aggregate energy balances because of power grid capacities. Due to a number of problems to be addressed to make public transport development able to cope with all the demands of the users, the issue of tram safety in cases of collisions remained neglected for a long time. Developers started to focus on this issue as early as in the late 1970s, but it took more than 30 further years to implement the concepts in practice. (See Figure 3 for the development of tram windshields.) During the development of these windshields, there is an apparent effort to address safety in the cases of collisions between two trams, a tram and a car, and a tram and a pedestrian.

Figure 3 shows that a breakthrough in the resolution of tram frontal safety emerged in the first decade of the 21st century. At that time, the protruding coupler was removed, and the windshields of trams started to be inclined to create deformation zones for tram–tram and tram–car collisions and to effectively enlarge the time of primary contact with a pedestrian in the case of a tram–pedestrian collision.

Currently, a vast majority of trams are fully or at least partially equipped with a low floor in the area for travellers. Two significant concepts can be distinguished in the scope of current tram production, as shown in Figure 4.

Both concepts have fronts that are similarly inclined with a protruding bumper. A difference can be seen in the positioning of the underframe. All trams with no exception are equipped with a so-called broom (Figure 5). The broom is a board that prevents any obstacle from going under the wheels.

In a tram with a long front overhang, the broom does not necessarily have to prevent contact with a pedestrian. However, this function is very important in the case of a tram with a short overhang. The broom is usually straight or an obtuse chevron, and with respect to the tram’s forward direction movement, it is usually inclined at an angle between 10 and 15°. A broom designed in this way effectively pushes obstacles from the track. Some constructions decrease the height of the broom above the ground in the case of frontal load to more effectively prevent the entry of an obstacle below the wheels and the underframe.

Especially noteworthy are the changes in the location and character of the primary impact, which determines the level of risk of serious life-threatening injuries. Figure 6 shows this change in the example of trams developed in Czechia from the 1950s to the beginning of the 2000s. To show it more clearly, there is a dummy in front of each front face (dummy—75% male).

In the 1950s, the impacts were primarily centred on the hips and only then on the entire body. In the counter motion, the head would strike the convex metal rim under the windshield. The protruding coupler was located in the middle of the tram.

The following decade saw the primary impacts occurring at the hip and shoulders nearly at the same time. As in the previous case, the head struck the convex metal rim under the windshield in the counter motion. The protruding coupler was still present. 

The first efforts to address tram frontal safety for pedestrians were made in the 1980s. The primary contact point was in the area of the thigh and then around the shoulders. The damage to the bodywork was reduced, and the head would thus strike the flat sheet metal rim under the windshield in the counter movement. Moreover, the cabin was significantly narrowed in the forward direction. This was not only because of the prescribed passable cross section of the track, but also due to an effort to increase the probability of throwing a knocked-down pedestrian off the track. The protruding coupler was, however, still present. 

The second half of the 2000s brought a principal breakthrough in the design of tram fronts. The protruding coupler was removed, and the profile of the bow was entirely inclined in order to elevate the hit pedestrian, thus dampening the pedestrian’s downward velocity. As a result, the risk of serious injuries, mainly those of the head, was reduced. The primary impact zone is directed to the thigh and then to the body, which is pressed as a whole against the tram’s front face. The head then strikes the lower part of the windshield in the counter movement.

## 3. The Primary Contact Principles

Figure 7 shows why the inclined tram front represents a breakthrough design. It illustrates the principal analysis of the force effects of a tram on a pedestrian during the primary impact.

A comparison of the diagrams above reveals that trams from the 1980s create, in conjunction with frictional forces between the sole of the pedestrian’s shoes and the ground, a tilting momentum. This momentum results in the pedestrian’s accelerated fall. Essentially, the pedestrian is pushed to the ground due to a negatively inclined front (i.e., against the travel direction), and the frictional forces between the soles and ground increase. In contrast, the tram front from the beginning of the 21st century, which inclined in the travel direction (i.e., positively), tended to elevate the pedestrian during the primary impact and thus reduce the friction between the pedestrian’s soles and the ground. The fall of the pedestrian to the ground was thus less accelerated. However, because the primary impact was still directed into the area of the thigh, the initial tilting momentum remained very similar. The detail of the head impact is also important. In the case of trams from the 1980s, the head hits the sheet metal under the windshield, and with a tram from the beginning of the 21st century, the head strikes the glass. Regarding the construction of both components, it is possible to adjust their rigidity and thus influence the severity of a pedestrian’s head injury. However, the windshield offers a much wider space thanks to its layered structure.

The current concepts of tram front design come out from the principles described above. Furthermore, the concepts entail a further mitigation of the devastating effects of collisions with pedestrians. The aim is to lower the site of primary contact and prepare a space for the subsequent fall of the head and body on the tram’s front so that the contact would be as soft as possible, and at the same time, the maximum amount of impact energy would be absorbed. The depiction and justification for the interpretation above are presented in Figure 8.

The concept, according to Figure 8, directs the force to the area of the forelegs. Then, the body is kept on a significantly inclined lower part of the tram’s front, and finally, the head strikes the lower part of the windshield. 

Obviously, the development favours fronts with a programmable rigidity of components that a pedestrian may strike. As a high level of impact energy has to be absorbed to cause deformations or destruction, the impact energy dissipates in these deformations or destruction. At the same time, there is an effort to elevate the pedestrian from the ground to make the pedestrian move together with the braking tram immediately after the impact. Due to both of those mechanisms, the acceleration of the pedestrian’s fall is minimised, thus reducing the severity of this fall.

A theoretical description of the above-discussed phenomena can be devised from the diagram shown in Figure 9.

A tram moving with velocity $V_{T}$ hits a pedestrian with force $F_{T}$ at height $L_{T}$. Assuming that the tram travels at velocity $V_{T}$ and the pedestrian has mass *m* related to their body size over density ρ(y),
(1)$$m=\int_{V}\rho (y)dV=\int_{0}^{W}dx\int_{0}^{H}\rho (y)dy\int_{0}^{D}dz=WD\int_{0}^{H}\rho (y)dy$$

The body is modelled as a cuboid, where *W*, *H*, and *D* are the width, height, and depth of the body. For a more accurate model of the body, a non-homogenous density of the cuboid is assumed so that the position of the centre of gravity is in the height as follows:(2)$$L_{M}=\frac{\int_{V}y\rho (y)dV}{m}=kH\mathrm{for}k\in (0.55÷0.57)$$

We established a coefficient of the site of the centre of gravity *k.* (Lower values are calculated for women.) The centre of gravity is moved above half of the height *H.* Furthermore, it is assumed that the mass of the body is concentrated in the centre of gravity, which moves in a plane (*x,y*) with velocity $v(t)=(v_{x}(t),v_{y}(t),0)$. At the moment of impact with an unknown force $F_{T}$ at the point of impact *C,* the body falls on the front, and the tram pushes the body forward. This force strongly depends on the deformation capabilities of the figure as well as of the tram’s front. It is difficult to determine this force. Our aim is to find out under what conditions the pedestrian will become elevated and thrown off the track instead of being brought under the wheels. The velocity of the centre of gravity $v(t)=(v_{x}(t),v_{y}(t),0)$ depends on the force $F_{T}$ as well as the friction force $F_{f}$ and their position relative to the centre of gravity of the pedestrian. In a general case, the body rotates (becomes inclined). The figure dynamics are established from the momentum balance (law of conservation of the angular momentum). The pedestrian is modelled as a solid of volume V. This solid is in contact with the tram at point *C*, with the location $x_{0}=\left(0,L_{T},0\right)$. The magnitude of angular momentum is calculated at this point. The general form of this law is as follows: (3)$$\overset{.}{\bar{\underset{\mathrm{angular}\;\mathrm{momentum}\;\mathrm{of}\;\mathrm{inertia}}{\underbrace{\int_{V}\left(x-x_{0}\right)\times \rho vdV}}}}=\underset{\mathrm{angular}\;\mathrm{momentum}\;\mathrm{of}\;\mathrm{contact}\;\mathrm{forces}}{\underbrace{\int_{\partial V}\left(x-x_{0}\right)\times tda}}+\underset{\mathrm{angular}\;\mathrm{momentum}\;\mathrm{of}\;\mathrm{volume}\;\mathrm{forces}}{\underbrace{\int_{V}\left(x-x_{0}\right)\times \rho fdV,}}$$
where “×” means the cross products of two vectors, and $\dot{v}(t)=({\dot{v}}_{x}(t),{\dot{v}}_{y}(t),0)$ is the acceleration in the corresponding direction ($\dot{x}(t)$—the time derivative of the function x(t)). This relation expresses that a time change in the angular momentum of a pedestrian is caused by moments of surface (contact) and volume forces. Assuming that
(4)$$x-x_{0}=\left(x,y-L_{T}\right),f=\left(0,g,0\right),F_{f}=\left(0,f_{E}mg,0\right),\dot{x}-{\dot{x}}_{0}=\left(v_{x},v_{y},0\right),$$
the friction coefficient between the pedestrian and the ground is $f_{E}\in (1,0.05)$, where low values are valid for very slippery surfaces, e.g., smooth ice. 

The angular momentum of inertia of the pedestrian is established as follows: (5)$$\begin{array}{l} \int_{V}\left(x-x_{0}\right)\times \rho vdV=\int_{V}\left(x,y-L_{T},0\right)\times \rho \left(v_{x},v_{y},0\right)dV=\\ \int_{V}x\rho \left(y\right)v_{y}dV-\int_{V}\left(y-L_{T}\right)\rho \left(y\right)v_{x}dV\;\;\;\;\;\;\mathrm{only}\;\mathrm{the}\;z\;\mathrm{component}\;\mathrm{is}\;\mathrm{nonzero} \end{array}$$

It is assumed that the velocity of the centre of gravity $v(t)=(v_{x}(t),v_{y}(t),0)$ depends only on time. For integration, definitions of the weight (1) and the centre of gravity (2) are used to derive the following:(6)$$\begin{array}{c} \int_{V=WHD}x\rho \left(y\right)v_{y}dV-\int_{V=WHD}\left(y-L_{T}\right)\rho \left(y\right)v_{x}dV=\\ v_{y}\int_{W}xdx\int_{D}dz\int_{H}\rho \left(y\right)dy-mL_{M}v_{x}+mL_{T}v_{x}=\frac{W}{2}mv_{y}-mL_{M}v_{x}+mL_{T}v_{x} \end{array}$$

Based on these simplifying assumptions, the magnitudes of the individual angular momentums in Equation (3) are as follows:

The time change of the angular momentum of inertia of the pedestrian equals
(7)$$\frac{W}{2}m{\dot{v}}_{y}-m{\dot{v}}_{x}\left(L_{M}-L_{T}\right)$$

The gravity acts by the angular momentum are as follows:(8)$$\int_{V=WHD}x\rho \left(y\right)gdV=\frac{W}{2}mg$$

The only surface force is the friction force $F_{f}$, whose angular momentum is
(9)$$\int_{HD}{\left.\left(x-x_{0}\right)\right|}_{y=0}\times tda=L_{M}f_{E}mg$$

By substituting the relationship between the acceleration of the centre of gravity of the pedestrian in the direction of travel and that in the perpendicular direction into the equation of the total angular momentum balance (3), the following is derived:(10)$${\dot{v}}_{y}-g=\frac{2}{W}\left[f_{E}g+(L_{M}-L_{T}){\dot{v}}_{x}\right]$$

This can be adjusted using the relation for the location of the centre of gravity (2):(11)$$\frac{{\dot{v}}_{y}}{g}-1=\frac{2kH}{W}\left[f_{E}+\left(1-\frac{L_{T}}{kH}\right)\frac{{\dot{v}}_{x}}{g}\right]>0$$

The condition for lifting the pedestrian above the ground with a subsequent displacement is derived below.
(12)$$f_{E}>\left(\frac{L_{T}}{kH}-1\right)\frac{{\dot{v}}_{x}}{g},\mathrm{for}kH>L_{T}$$

The result is a condition for tram front construction: the primary impact always has to happen at a lower point than the pedestrian’s centre of gravity.

Whether the pedestrian slides down on the ground again or not from the tram’s front is to be determined. After the attenuation of the dynamic part of the fall, this depends mainly on the size of the angle of inclination of the tram’s front α and the friction coefficient $f_{T}\in \left(1.4,0.6\right)$ between the pedestrian and the tram’s surface (Figure 9). A higher value holds for a suitable refined (shapes) surface, which increases friction. The value of 0.6 runs for the contact of leather with metal in the case of a smooth surface.

The friction force on the front part of the tram and the force pushing the body to the ground equal the following (see Figure 9):(13)$$F_{T,fric}=f_{T}mg\cos\alpha$$
(14)$$F_{low}=mg\sin\alpha$$

A suitable angle of the tram front follows from the conditions below: (15)$$F_{T,fric}\geq F_{low},\;\mathrm{or}\;\mathrm{tg}\alpha \geq \frac{1}{f_{T}},\alpha \in (32{}^{\circ},75{}^{\circ})$$

Under these conditions, the body will be displaced by the tram. This model is an oversimplification, and the final result will depend not only on the friction but also on the mechanics of the body’s fall accompanied by potential subsequent repercussions. In practice, the angle of the lower part of the tram approaches 30°.

## 4. Other Context

A unified and generally accepted methodology to test the safety of rail vehicles does not currently exist. Nevertheless, this article shows that the conditions for the development of tram front face designs are so specific in the issue of their safety in collisions with pedestrians that an effective resolution has actually been defined. The development proceeded in an unprompted manner in the vast majority of manufacturers. Furthermore, the technical report, CEN/TR 17420:2020 [27], is based on this fact. This report defines the basic parameters and characteristics of tram fronts with the aim of decreasing the consequences of collisions with pedestrians. Among other recommendations, this report contains requirements for tram design shaping.

The main shape criteria are as follows:

(1) Slope of the front in the longitudinal direction and its “jaggedness” 

This parameter is essential for the impact wave and decomposition of force effects on the pedestrian that has been hit. Figure 8 indicates the requirements.

To further elucidate what was described above, line B is plotted into the contours of the older trams’ front faces in Figure 10. The height of the foremost point of the tram is also dimensioned (in accordance with Figure 11).

Obviously, more recent tram designs (since the beginning of the 21st century) already have a tendency towards smoother frontal shapes and a reduction in the height of the foremost frontal line to suppress the extent of the negative slope.

(2) Front floor plan line

The front floor plan line is another important parameter with a significant impact on the character of consequences of collisions with pedestrians. In the technical report, CEN/TR 17420:2020 [27], the following geometric requirements (TW = maximum vehicle width) were established for the foremost points of the front floor plan line (corresponding with the height, hs), as displayed in Figure 12.

In Figure 13, the defined frontal floor plan line according to CEN/TR 17420:2020 [27] is plotted into the floor plan of a tram from the 1980s. It is evident that the new requirements of the technical report for the floor plan line of the foremost points of the tram’s front face have established a smooth transition without any sharp shape changes.

Although the above discussion was based on the trams used in Prague, a similar trend can be observed in other European cities such as Berlin. Berlin’s fleet has been modernised using low-floor trams since the 1990s (Figure 14).

Even in the case of these trams, a progressive spontaneous development of their fronts is evidently in line with the CEN/TR 17420:2020 report [27].

Generally, the development of the fronts of rail vehicles in terms of their safety in the cases of collisions lags behind that in the car industry. This is partly because rail tracks operate in dedicated areas; thus, the described issues were not so pressing for a long time. Another reason is that developers had to address a number of other tasks to ensure functional, effective, and reliable transportation. However, trams represent a highly specific type of rail vehicle because trams are not operated in a prevailingly separate, dedicated area like trains. In fact, trams are exposed to complex conditions of heavy traffic in urban agglomerations that they have to deal with.

Based on the long-term statistics of tram–pedestrian collisions from the files from Prague’s municipal transport authority (more than 65 years), the most frequently affected impact points were selected. In countries with left-hand driving, the most frequently affected impact point is the right front part of tram fronts. New progressive development in this area came with lowering the edge of windshields so that the pedestrian´s head strikes the glass area of the front face in a collision. The windshield is becoming a crucial part of the deformation zone of the tram and defines the response of the front face to the impact. The layered material of the windshield offers wide possibilities to meet the requirements on the simple programming of rigidity of the windshield and enables the dissipation of a significant amount of impact energy during the windshield’s destruction. Furthermore, a general effort is made to decrease the sites of the primary impact to the area under the knees and to elevate a knocked-down pedestrian immediately after the impact. Depending on the tram design (with long or short overhang), there are varying solutions to prevent the pedestrian from being knocked over.

Understanding the reasons for the development of tram fronts opened the possibility of participation for other bodies and organisations besides the tram manufacturers. At the EU level, a discussion on creating a legislative framework is in progress. Such a framework would make the field of tram windshield development level out with that of car windshields. To reach this aim, it will be necessary to set computational models of simulations of impacts and a methodology for physical crash tests.

## 5. Experimental Work

As the head is the most frequent body part involved in tram–pedestrian collisions, this study focused on the head injury assessment using the head injury criterion (*HIC*_15_). Four different tram types, typical for Prague’s public transport, were included. The crash tests were conducted at four different speeds of the approaching tram (5 km/h, 10 km/h, 15 km/h, and 20 km/h). There were two crash tests performed at each speed: one for a frontal impact and one for a side impact.

### 5.1. Tram Types Used

Two older tram types that are still in use in Prague, T3R.PLF (Figure 15—left) and KT8D5 (Figure 15—right) (both ČKD Tatra, Prague, Czech Republic), and two modern ones, 14T (Figure 16—left) and 15T (Figure 16—right) (both Škoda Transportation, Pilsen, Czech Republic), were used. A total number of 32 collisions were conducted and analysed (one collision for each speed, tram type, and position of an ATD) using a 200 m straight tram track at the testing facilities of the VÚKV a.s. (Research Institute for Railway Vehicles). Each collision was conducted only once; therefore, a thorough statistical analysis could not be conducted, which is a significant limitation of this study.

### 5.2. Used Dummy—Anthropomorphic Test Device (ATD)

A Hybrid III 50th Percentile Male Pedestrian Dummy (JASTI, Tokyo, Japan) was used to simulate the biomechanical response of a pedestrian to the impact of a tram. The dummy was equipped with internal and external body sensors to measure the kinematics and dynamics of a collision. The head, chest, and pelvis were equipped with wireless accelerometers and gyroscopes, and the thighs, calves, and several spinal segments (C1, C7, TH5, L5, and S5) were equipped with wireless dynamometers (Figure 17—left). The data from the in-body sensors were recorded at the frequency of 20,000 Hz. The kinematics of the external sensors (reflective passive markers) was recorded using the Qualisys motion capture system (Qualisys AB, Göteborg, Sweden). The external sensors were attached to the dummy’s head, torso, thighs, and calves and to the front end of a tram (Figure 17—right). 

### 5.3. Instrumentation and Collision Site

Two sets of four cameras were used to record the kinematics of both the primary impact (the site of the initial contact) and the whole tram–pedestrian complex (Figure 18). The recording frequency of the motion capture system was 300 Hz. 

The movement analysis of the collision itself and the whole tram–pedestrian system was conducted using two ultra-high-speed cameras (Photron, Tokyo, Japan). One camera recorded the site of the initial contact (with a recording frequency of 12,000 Hz), and the other one recorded the whole tram–pedestrian complex (with a recording frequency of 500 Hz) (Figure 18).

### 5.4. Collision Scenarios

The frontal and side impact scenarios followed the recommendations of the current version of the technical report by CEN (CEN/TR 17420) regarding the position of the ATD on the tram track [26]. In both scenarios, the ATD was placed at a distance equal to 15% of half of the tram’s width from the centre line towards one end of the track (Figure 19). Although the technical report only considers a case of side impact (a pedestrian facing the tram sideways), the current study included the frontal impact as well, mainly because the ATD is only validated for frontal crash tests. However, the verification of an ATD for side-impact collisions has already been underway as another part and goal of the whole project.

### 5.5. Data Collection, Processing, and Analysis

The kinematic data were recorded and analysed using the Qualisys motion capture system, ultra-high-speed cameras, and the tram’s tachograph. The dynamic data were assessed using the in-body sensors, especially the accelerometers. Three-axis accelerometers were used to measure the accelerations that took place in relation to all three Cartesian coordinate axes. The primary outcome variable was the resultant head acceleration (measured in multiples of the acceleration of gravity, *g*) during the primary impact:(16)$$\left|\vec{a}\right|=\sqrt{a_{x}^{2}+a_{y}^{2}+a_{z}^{2}}$$
where ***a*** represents the resultant head acceleration (g), and ***a_x_***, ***a_y_***, and ***a_z_*** represent the accelerations of the head in each direction (g).

To assess the risk and severity of a given head injury, the head injury criterion (*HIC*_15_) was used. This parameter describes the likelihood of a head injury due to impact or acceleration, and it can be estimated by integrating the resultant head acceleration over a given time interval as follows [26,28]:(17)$${HIC}_{15}=\left\{\left(t_{2}-t_{1}\right)\cdot {\left[\frac{\int_{t_{1}}^{t_{2}}\hat{a}(t)dt}{\left(t_{2}-t_{1}\right)}\right]}^{2.5}\right\}max$$
where $\hat{a}$(***t***) represents the normalised head acceleration measured, $\hat{a}=\frac{a}{g}$, where ***a*** is the resultant head acceleration and ***g*** is the gravity acceleration. Therefore, the normalised head acceleration is unitless; ***t*_1_** represents the start of the time interval (s), ***t*_2_** represents the end of the time interval (s), and (***t*_2_**–***t*****_1_**) represents the width of a given time interval during which the ***HIC*** assumes the maximum value (s). 

The National Highway Traffic Safety Administration (NHTSA, Washington, DC, USA) adopted the limits that reduced the maximum time interval for estimating the ***HIC*** to 15 ms (*HIC*_15_) because relatively short-duration impact waveforms are considered to be associated with a head impact [29,30]. 

To assess the severity of a given injury under certain test conditions, the value of the *HIC* can be linked to an existing level of injury on the Abbreviated Injury Scale (AIS) [31]. The scale ranges from AIS0 (“no injury”) to AIS6 (“potentially fatal injury”), and the *HIC* has been tested and validated under a variety of conditions to be converted to the probability of a given injury, as defined by the AIS code (Table 1) [31,32].

The most advanced legislation as well as crash test providers (e.g., Euro NCAP, the European New Car Assessment Programme) or NHTSA (USA)) have been mostly focused on determining the probability of AIS3+ (i.e., AIS ≥ 3) or AIS4+ (i.e., AIS ≥ 4) injuries, corresponding to serious/severe injuries or worse, in terms of the HIC values [28,31,34]. Converting the *HIC*_15_ value into an estimate of AIS3+ or AIS4+ injury can be carried out using correlation equations based on the cadaver experiments by Prasad and Mertz [35] and expressed in a sigmoid form (Figure 20).
(18)$$p(AIS3+)=\frac{1}{1+e^{\left[(3.39+140/{HIC}_{15})-0.00531\cdot {HIC}_{15}\right]}}$$
(19)$$(AIS4+)=\frac{1}{1+e^{\left[(4.90+140/{HIC}_{15})-0.00501\cdot {HIC}_{15}\right]}}$$
where ***p*(*AIS3+*)** represents the probability of injury corresponding to a code of three or higher on the Abbreviated Injury Scale based on the *HIC*_15_ (18), and ***p*(*AIS4+*)** represents the probability of injury corresponding to a code of four or higher on the Abbreviated Injury Scale based on the *HIC*_15_ values (19).

## 6. Results

The analysis was focused on the assessment of resultant head acceleration and the severity of a possible injury using the *HIC*_15_ value and corresponding injury level on the AIS during the primary impact. The probability of injury corresponding to a code of three or higher, ***p*(*AIS3+*)**, and equal to or higher than four, ***p*(*AIS4+*)**, on the Abbreviated Injury Scale were calculated. The primary impact was defined by the initial contact between the ATD and tram, and it ended with the ATD being thrown forward by the tram (Figure 21). 

### 6.1. Frontal Impact

Regarding the frontal impact, the video analysis showed that the initial contact was specific for each tram type (Figure 22, Figure 23, Figure 24 and Figure 25). In the case of the oldest type, T3R.PLF (Figure 22), the initial contact was almost simultaneous; the initial points of contact were the hip/thigh area of the ATD with the bumper and the head of the ATD with the hard part of the tram’s front. This was very similar to the other older type, KT8D5 (Figure 23). However, in this case, the initial contact was made to the head of the ATD with the upper part of the tram’s front, just below the windscreen, followed by an impact into the hip/thigh area. As a consequence of a frontal collision with both of these tram types, the ATD was thrown violently forward, resulting in a severe secondary impact on the surrounding infrastructure (e.g., pavement, track, and side poles). 

In the case of modern tram types, the contact with the head of the ATD always followed the initial contact with the hip/thigh area (14T type) or knee area (15T). A frontal collision with the 14T tram type resulted in the head hitting the bottom part of the tram’s windscreen (Figure 24). Momentarily after the collision, the ATD remained in contact with the tram before being thrown forward. In the case of the 15T tram type, the initial impact was at the knee area of the ATD, immediately followed by the hip/thigh area. The ATD then hit the windscreen with its head, and similarly to the 14T type, the ATD and tram stuck together after the primary impact (Figure 25). The ATD was then thrown forward. In the case of the 15T tram type, the windscreen sustained damage from the head impact at a speed of 20 km/h. 

Regarding the primary impact phase, the head acceleration data showed an increasing trend towards higher values with an increasing speed for all four tram types (Table 2 and Table 3 and Figure 26). The corresponding *HIC*_15_ values showed a similar increasing trend with an increasing speed (Table 2 and Table 3 and Figure 27). In the case of the 15T tram type, though, the values were considerably higher for each speed up to 20 km/h. At that point, the windscreen endured significant damage and cracked only with the impact of the ATD’s head. Furthermore, a slight drop towards a lower value of *HIC*_15_ was recorded. However, the *HIC*_15_ values for all tram types and speeds did not exceed the value of 300, and the probability of AIS3+ or AIS4+ injuries fell within a range of 10%.

### 6.2. Side Impact

Regarding the side impact, the video analysis showed that the initial contact between the ATD and the tram varied depending on the tram type (Figure 28, Figure 29, Figure 30 and Figure 31). In the case of the oldest type, T3R.PLF (Figure 28), the first contact was between the bumper and the hip/thigh area of the ATD, followed by the shoulder striking a hard part of the tram’s front. The initial contact between the other older type, KT8D5 (Figure 29), and the tram was almost simultaneous; firstly, the tram hit the shoulder of the ATD, and then the upper part of the tram’s front struck the hip/thigh area. There was no contact with the head of the ATD in any of the collisions. The ATD was then propelled forward, resulting in a hard secondary impact on the surrounding infrastructure. 

In the case of modern tram types, the tram first hit the hip/thigh area of the ATD (14T) or the knee area (15T), followed by contact with the arm (14T) or hip/thigh area (15T). In the case of the 14T, contact between the tram and the head of the ATD (Figure 30) was not observed. In the case of the 15T tram type, the initial contact was made to the knee area of the ATD, immediately followed by the hip/thigh area (Figure 30). The ATD only hit the windscreen with its head at higher speeds (15 km/h and 20 km/h), with its feet leaving the ground, too. 

Regarding the primary impact phase, the head acceleration data showed an increasing trend towards higher values with an increasing speed for all four tram types (Table 4 and Table 5 and Figure 32). The corresponding *HIC*_15_ values showed a similar increasing trend with an increasing speed (Table 3 and Table 4 and Figure 33). In the case of the 15T tram type, though, the values were considerably higher for the speeds of 15 km/h and 20 km/h. 

## 7. Discussion

This study examined two possible scenarios of a tram–pedestrian collision, the frontal and side impacts, with four tram types that are typical for Prague’s public transport system approaching the site of collision at four different speeds (5 km/h, 10 km/h, 15 km/h, and 20 km/h). A total of 32 collisions were conducted, including one for each scenario, tram type, and speed, using an anthropomorphic test device, i.e., a crash test dummy. The analysis was focused on the resultant head acceleration of the primary impact as well the assessment of the risk and severity of a possible given head injury using a head injury criterion (*HIC*_15_). The *HIC*_15_ was then linked to a level of injury on the Abbreviated Injury Scale (AIS). 

### 7.1. The Frontal Collision

Regarding the frontal collision and its primary impact phase, the results showed an increasing trend towards higher values of head acceleration with an increasing speed for all four tram types. These results are similar to those of other studies focused on the collisions of pedestrians and vehicles [36,37]. However, these studies only used multibody modelling and simulations with different designs of cars, and there is no such study involving trams, which is the main focus of the current study. On the other hand, the values of the resultant head acceleration fell within the range of 114–124 g, and the corresponding *HIC*_15_ did not exceed the value of 300 for any speed or tram type. Therefore, the results suggest that the frontal collision could result in mild or moderate head injuries at these tram speeds. An interesting finding of this part was in the case of the 15T tram type, where the *HIC*_15_ values were considerably higher for each speed until 20 km/h. At that moment, the windscreen sustained significant damage and cracked from the impact of the ATD’s head, and there was a slight drop towards a lower value of the *HIC*_15_. This result is similar to two other studies showing that head-to-windscreen impacts could reduce the risk of head injuries significantly [36,37]. The results therefore suggest that it is not just the vehicle impact velocity, but it could also be the location of initial contact and the properties of the windscreen influencing the risk and severity of head injuries during these collisions.

### 7.2. The Side Collision 

Regarding the side collision, the head acceleration data showed an increasing trend towards higher values with increasing speed for all four tram types. These results are similar to other studies focused on the collisions of pedestrians and vehicles [36,37]. Except the 15T tram type, the maximum head acceleration did not exceed 30 g for all other types. In the case of 15T, there was a trend towards higher values at the speeds of 15 km/h and 20 km/h, reaching maximums of 34.1 g and 83.4 g, respectively. At these speeds, the head of the ATD hit the windscreen, though without damaging it, which might account for these higher load values compared to the other tram types. 

The corresponding *HIC*_15_ values exhibited a similar increasing trend with an increasing speed. In the case of the 15T tram type, though, the values were considerably higher for speeds of 15 km/h and 20 km/h. However, the *HIC*_15_ values exceeded the value of 200 only for the 15T tram type (205.5), and regarding the other tram types, the *HIC*_15_ did not exceed the value of 35. Therefore, the probability of AIS3+ or AIS4+ injuries was less than 1%, and in the case of 15T, the probability did not exceed 5% (20 km/h). Unfortunately, there is an apparent significant gap in the scientific literature that does not permit a comparison with these results with similar studies. The ATS also presents limitations because it is not verified for the side impacts. However, the results suggest that during the primary impact phase, the side collision could result in no or very minor head injuries. 

### 7.3. Limitations and Future Directions

To our knowledge, this is the first study focused on the tram–pedestrian collisions using crash test experiments with an anthropomorphic test device and two different impact scenarios, the frontal and side impacts. However, there are several methodological considerations and limitations that need to be addressed when interpreting the results. Firstly, there was only one collision conducted for each scenario, tram type, and impact speed of a tram, which limited the subsequent, more thorough statistical analysis. Secondly, the type and orientation of the ATD during both collision scenarios was the same; the Hybrid III 50th Percentile Male Pedestrian Dummy stood with its feet hip-width apart and with its arms lowered along the body. The study did not involve any female or child anthropomorphic test devices or any specific walking phase at the time of the collisions. Thirdly, the used ATD is only verified for frontal impacts. Therefore, there is no evidence that this type of the ATD can be used for the side impacts. 

Future studies should therefore include more tram–pedestrian collisions using various ATDs in different orientations and impact scenarios. They could be either focused on the primary impact or secondary impact with the ground, and it could be beneficial to conduct studies assessing tram–pedestrian collisions using multibody modelling and simulations. Based on the results of the current study, the future studies could also include testing and analysing the use of collision avoidance systems and the influences of the windscreen and its properties on the primary impact.

There were two good reasons to use the Hybrid III 50th Percentile Male Pedestrian Dummy for the purposes of this study. Firstly, it is one of the most common and used crash test dummies in the world, and secondly, it was immediately available and provided by the Faculty of Physical Education and Sport. The research team is aware of the dummy being designed and utilised for slightly different purposes and forensic scenarios, but there has been a continuous development and demand for other applications (such as a pedestrian kit that was introduced recently). The design of the dummy’s shoulder joint is a source of the main differences between a dummy and a human regarding the biomechanical response to an impact, which can lead to inaccurate results, especially in the case of a side impact. There are several other ATD devices that could provide better results in this area (such as Polar by Honda or Primus by CTS&Kistler). The Primus dummy, a biofidelic crash test dummy, also exists in a breakable version with its material corresponding to the mechanical properties of human tissues. Therefore, the team is aware of other possibilities on how to further continue in this area of research and what the next steps could be. However, to provide a high-quality analysis of the vehicle’s safety, the reproducibility and standardisation of crash tests and experiments are key. Thus, in such cases, some degree of simplification is acceptable to provide thorough and complex data that can be used for the following statistical analysis. 

## 8. Conclusions

Both collision scenarios, frontal and side impacts, of tram–pedestrian accidents can result in very high values of resultant head acceleration and corresponding *HIC*_15_ values during the primary impact, especially in the frontal one. Although the *HIC* values can be quite low, the injuries diagnosed can be very serious. An essential aspect that has not been highlighted enough is the movement of the head in relation to the body. Thus, the severity of the injury is strongly affected by the extreme tilting and/or rotation of the head relative to the torso [38].

The outcomes of the study showed that tram–pedestrian collisions were very complex, and the risk and severity of a head injury could be strongly influenced by the position and orientation of the ATD at the time of the collision, the impact speed and design of the vehicle’s front end, the impact mechanism, and the location of initial contact. The results also showed significant differences between older (T3R.PLF and KT8D5) and modern tram types (14T and 15T). The modern types were found to be more likely to cause more serious head injuries than the old ones. The positive effect of lifting and rolling the ATD on the windscreen of tram was thus outweighed by the hard impact of its head on the windscreen, which exhibited an apparently higher stiffness compared to the sheet metal panels of the older models.

The essential result of the above-mentioned study is the confirmation of the hypothesis that the windscreen is an essential element of the vehicle’s passive safety for pedestrians in modern types of trams with an inclined front end and a low glass edge. It is therefore obvious that without a responsible approach to the research and development of the properties of windscreens, the CEN/TR 17420:2020 technical report cannot provide any desired improvements in the pedestrian’s safety.

Therefore, the next part of our series of three articles describes the results of testing currently used windscreens, and it proposes a methodology for how to analyse their mechanical properties and how to estimate their absorption capacity. This could lead to a better understanding and identification of glass parameters and their influences on life-threatening injuries to pedestrians, especially to their heads.

## Figures and Tables

**Figure 1 sensors-23-08819-f001:**
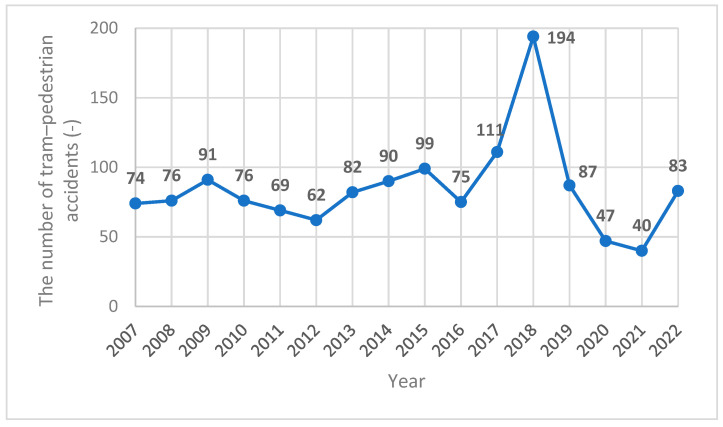
The number of tram–pedestrian accidents per year in Prague between 2007 and 2020.

**Figure 2 sensors-23-08819-f002:**
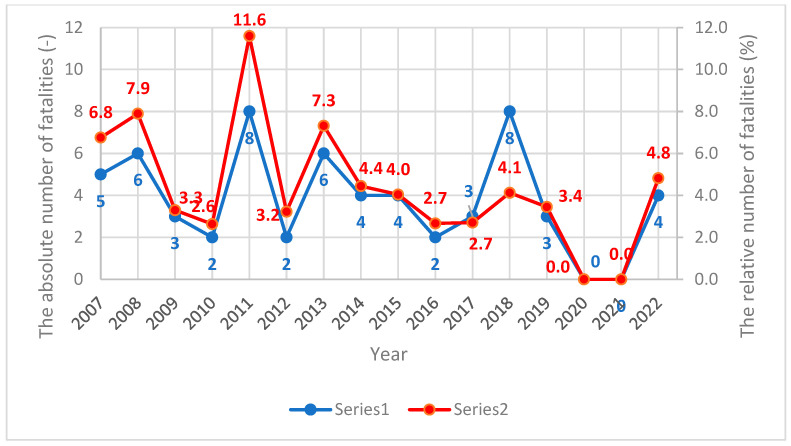
The absolute (blue line) and relative (red line) number of fatalities per year between 2007 and 2020.

**Figure 3 sensors-23-08819-f003:**
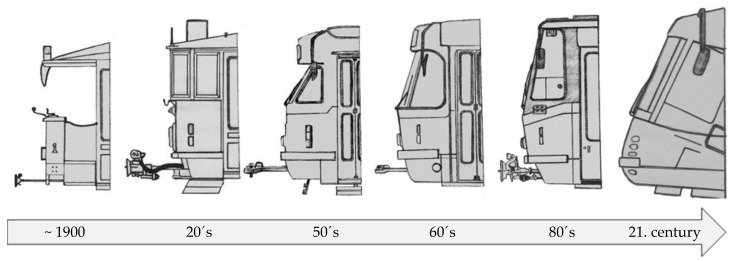
Development of tram windshields (author’s archive).

**Figure 4 sensors-23-08819-f004:**
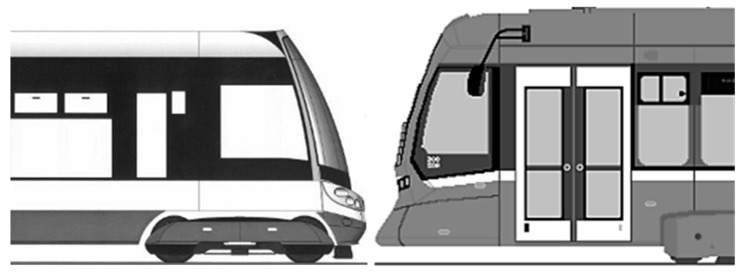
Two current concepts of the tram front face design (author’s archive).

**Figure 5 sensors-23-08819-f005:**
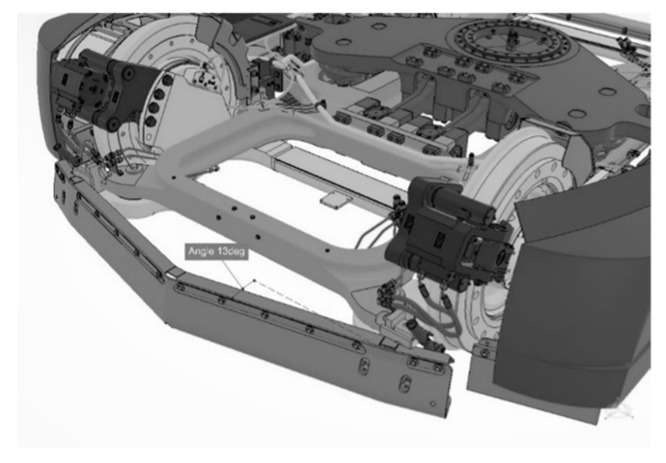
Broom design (left: straight; right: arrow) (author’s archive).

**Figure 6 sensors-23-08819-f006:**
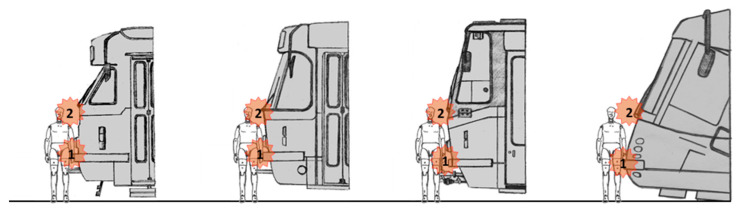
Development of the location of contact areas on tram front faces in the primary impact (1—initial contact; 2—head impact) (author’s archive).

**Figure 7 sensors-23-08819-f007:**
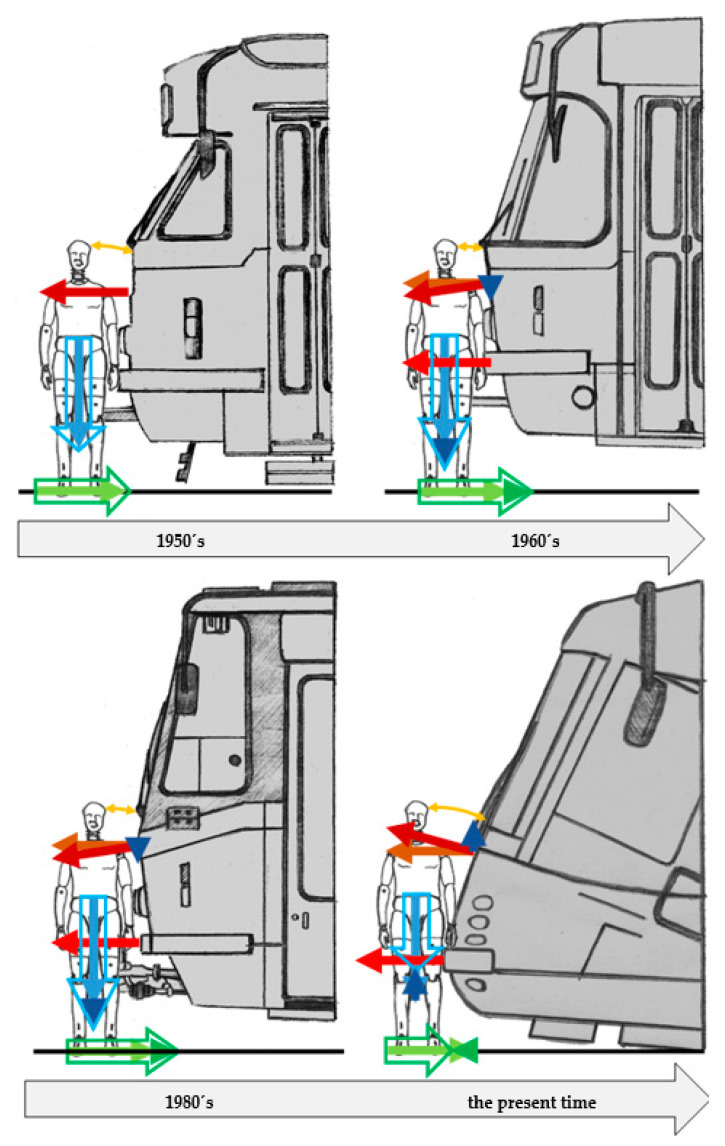
Development of force effects on tram fronts (author’s archive) (blue—vertical forces; red—contact forces between the pedestrian and the tram; green—frictional forces on the ground; empty arrows—resulting forces; orange—trajectory of the head from initial position to the contact with the tram).

**Figure 8 sensors-23-08819-f008:**
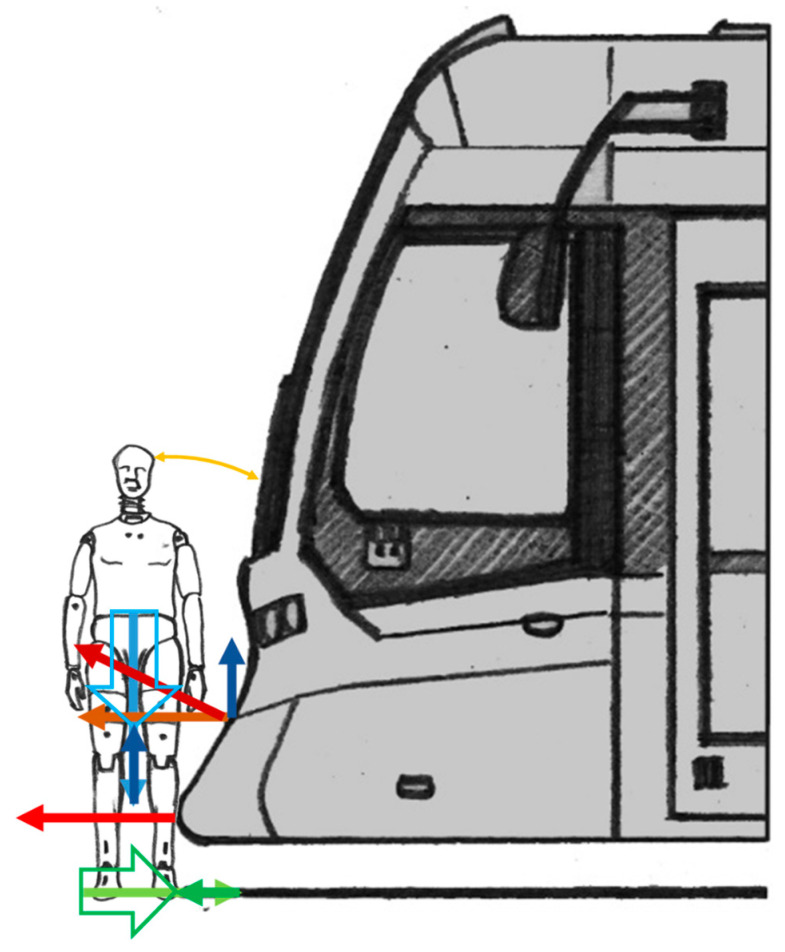
Primary impact in the case of the current design of tram fronts (author’s design) (blue—vertical forces; red—contact forces between the pedestrian and the tram; green—frictional forces on the ground; empty arrows—resulting forces; orange—trajectory of the head from initial position to the contact with the tram).

**Figure 9 sensors-23-08819-f009:**
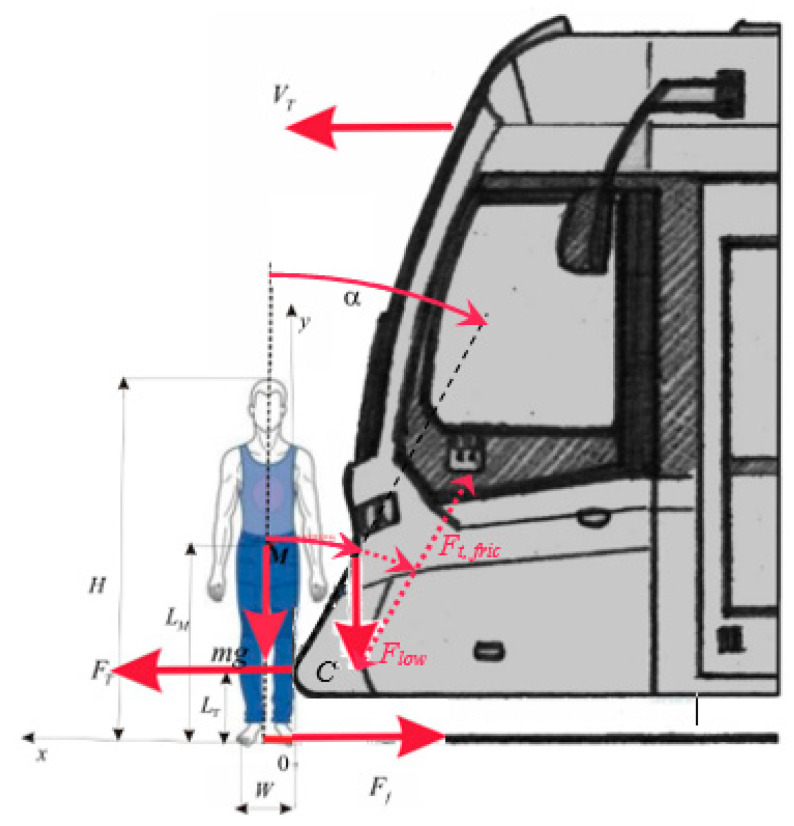
A diagram of tram–pedestrian collision.

**Figure 10 sensors-23-08819-f010:**
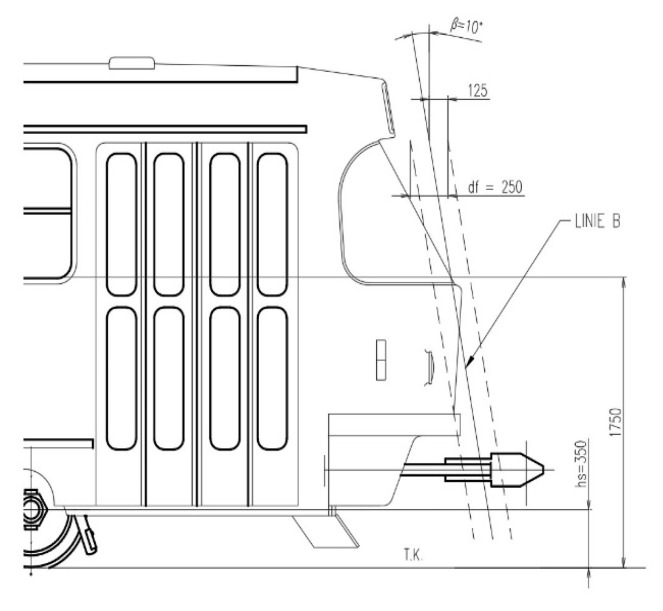
Line of inclination of the front according to CEN/TR 17420:2020 [27] in contrast to older tram types (author’s archive).

**Figure 11 sensors-23-08819-f011:**
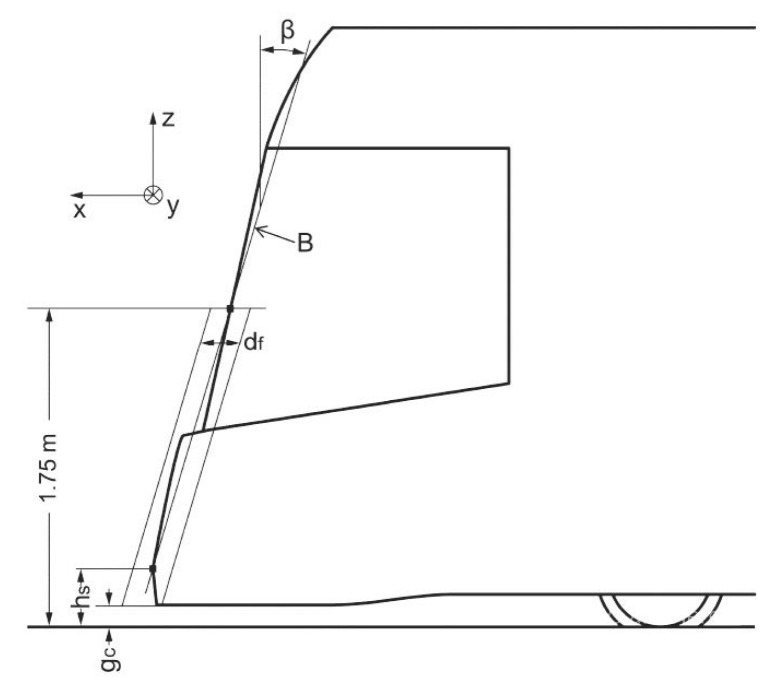
Longitudinal inclination of the front according to CEN/TR 17420:2020 [27]. ***h_s_*** is the foremost point of the vehicle floor plan, which should be at a maximum height of 350 mm from the top of the rail (T.K.); ***B*** is the line of inclination with the connecting point ***h_s_*** and the point on the front of the vehicle at a height of 1.75 m (height of the impact surface); ***β*** is the inclination angle of line ***B*** (***β*** ≥ 10°); and ***d_f_*** = 250 mm, which is the line parallel to line ***B*** in the distance of ***d_f_*/2**, which defines the collision contact zone; 30 mm < ***g_c_*** < 130 mm.

**Figure 12 sensors-23-08819-f012:**
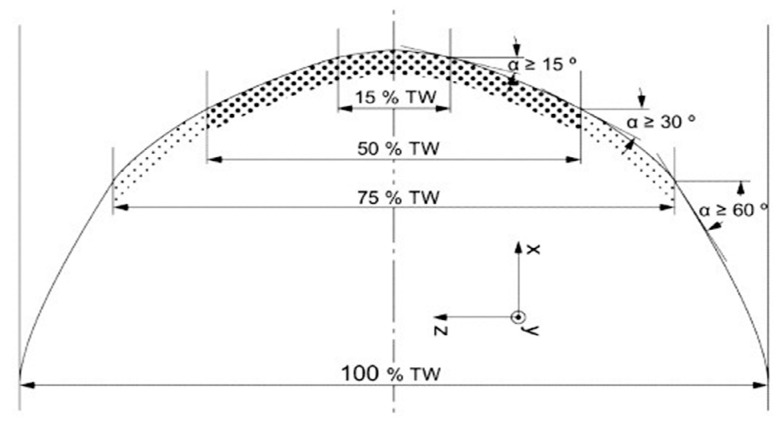
Front floor plan line according to CEN/TR 17420:2020 [27].

**Figure 13 sensors-23-08819-f013:**
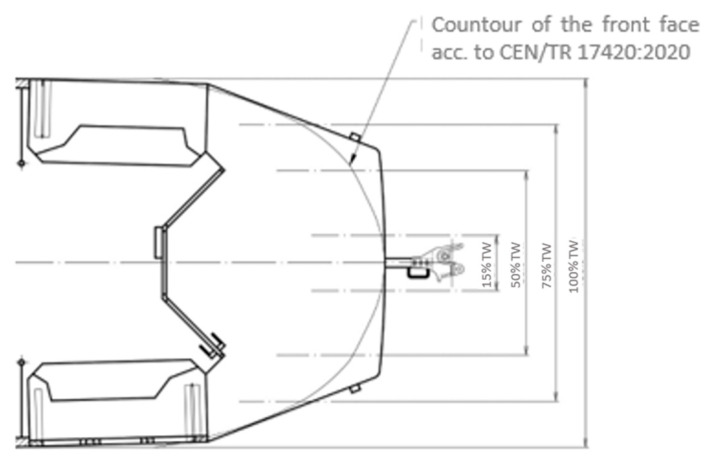
Floor plan line of front according to CEN/TR 17420:2020 [27] in contrast to floor plan of older trams (author’s archive).

**Figure 14 sensors-23-08819-f014:**
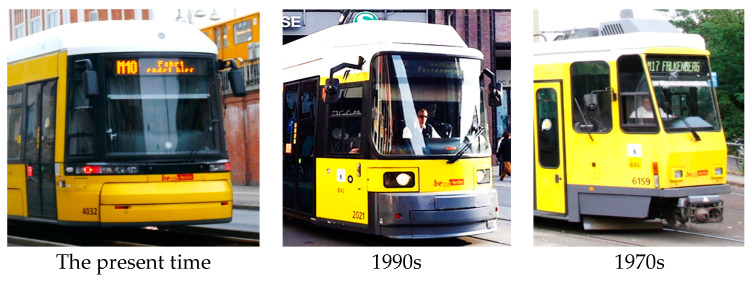
Berlin trams.

**Figure 15 sensors-23-08819-f015:**
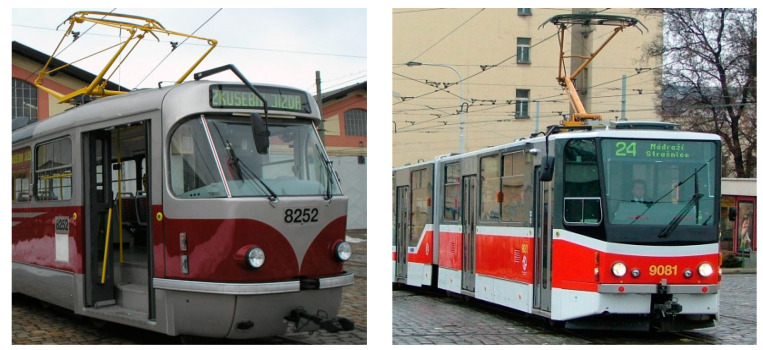
Older tram types used in tests (T3R.PLF type—**left**; KT8D5 type—**right**).

**Figure 16 sensors-23-08819-f016:**
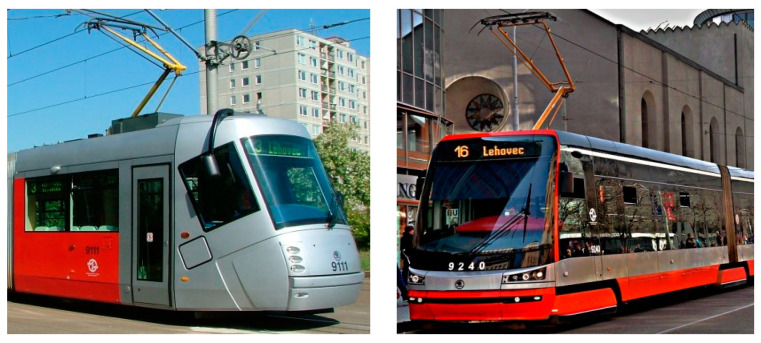
Modern tram types used in tests (14T type—**left**; 15T type—**right**).

**Figure 17 sensors-23-08819-f017:**
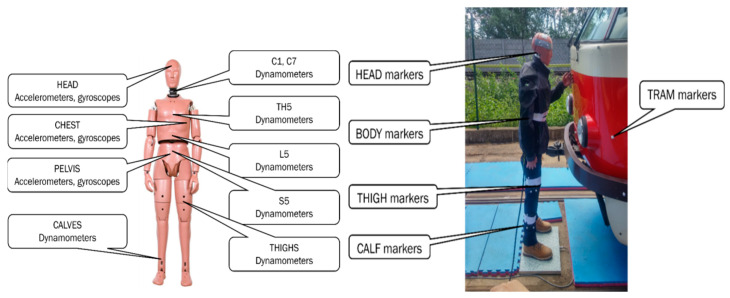
Internal (**left**) and external (**right**) body sensors used to record the kinematics and dynamics of an ATD and tram (markers on the front end of tram) during a collision.

**Figure 18 sensors-23-08819-f018:**
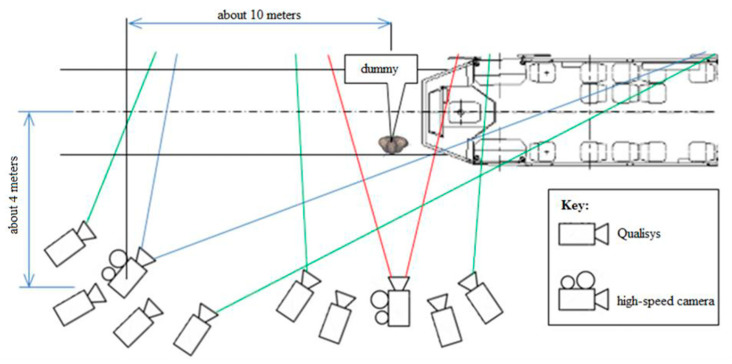
The configurations of both camera systems and their fields of view of the collision (green lines—the field of view of the Qualisys motion capture system; red lines—the field of view of the ultra-high-speed camera focused on the collision itself; blue lines—the field of view of the ultra-high-speed camera recording the whole tram–pedestrian complex).

**Figure 19 sensors-23-08819-f019:**
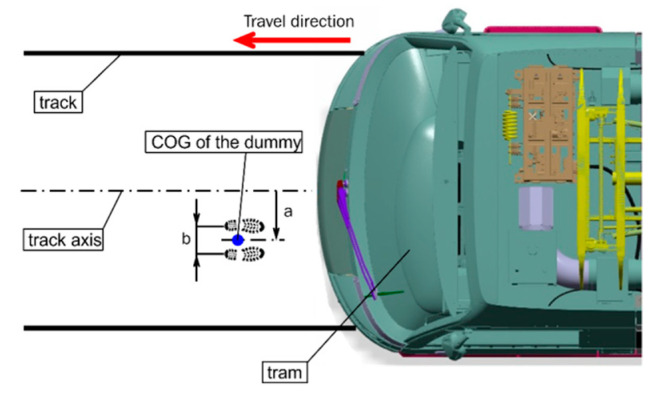
The position of an ATD on the tramway track in front of the approaching tram (the frontal impact—left; the side impact—right; a—the distance equal to 15% of half of the tram’s width from the centre line; b—the ATD’s standing position with feet hip-width apart).

**Figure 20 sensors-23-08819-f020:**
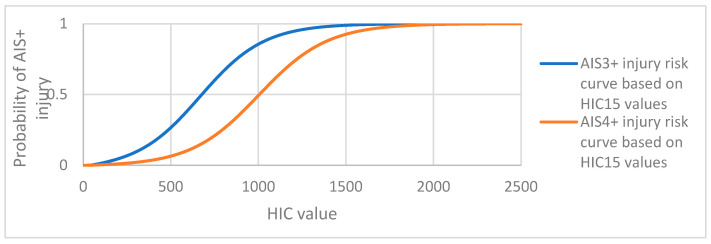
Head injury risk curves based on *HIC*_15_ values for AIS3+ and AIS4+ injuries.

**Figure 21 sensors-23-08819-f021:**
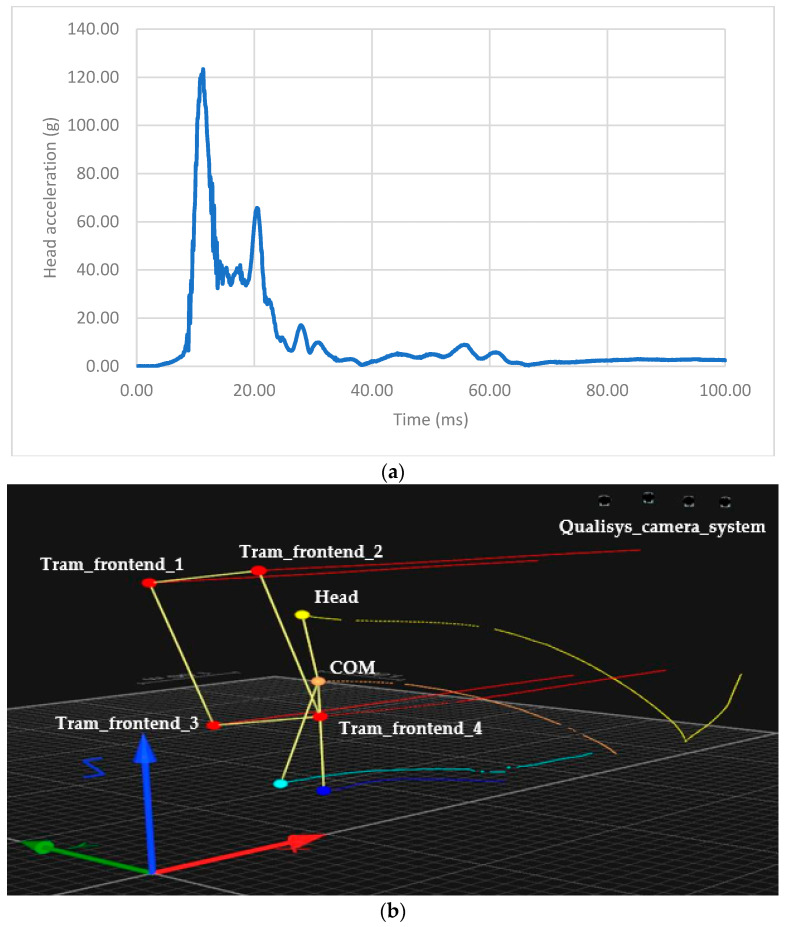
(**a**) An example of an analysis of a frontal collision between the ATD and T3R.PLF tram type at a speed of 20 km/h (the top—time evolution of head acceleration during the primary contact; the bottom—the collision based on the motion capture camera system). (**b**) An example of an analysis of a frontal collision between the ATD and T3R.PLF tram type at a speed of 20 km/h (the top—time evolution of head acceleration during the primary contact, flying phase, and the secondary impact; the bottom—the whole collision based on the motion capture camera system)—continuation.

**Figure 22 sensors-23-08819-f022:**
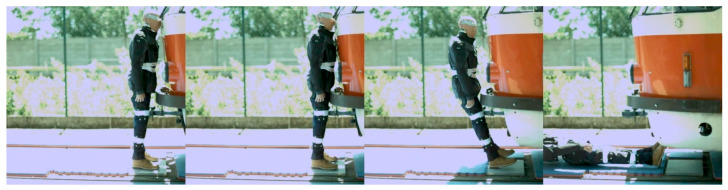
The frontal collision sequence between the ATD and T3R.PLF tram type approaching at a speed of 5 km/h.

**Figure 23 sensors-23-08819-f023:**
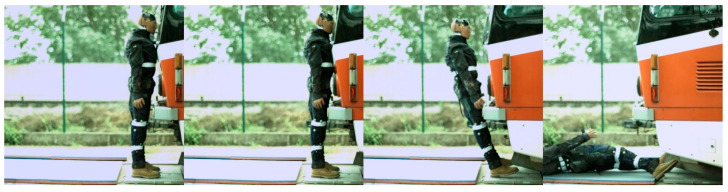
The frontal collision sequence between the ATD and KT8D5 tram type approaching at a speed of 5 km/h.

**Figure 24 sensors-23-08819-f024:**
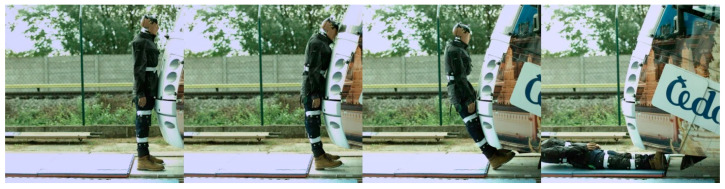
The frontal collision sequence between the ATD and 14T tram type approaching at a speed of 5 km/h.

**Figure 25 sensors-23-08819-f025:**
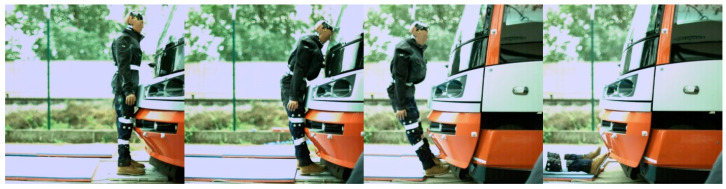
The frontal collision sequence between the ATD and 15T tram type approaching at a speed of 5 km/h.

**Figure 26 sensors-23-08819-f026:**
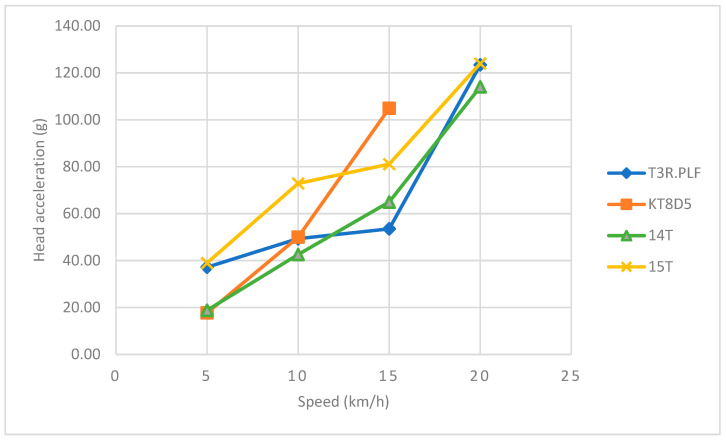
The influence of speed on the head acceleration for all four tram types during the primary impact in the frontal impact.

**Figure 27 sensors-23-08819-f027:**
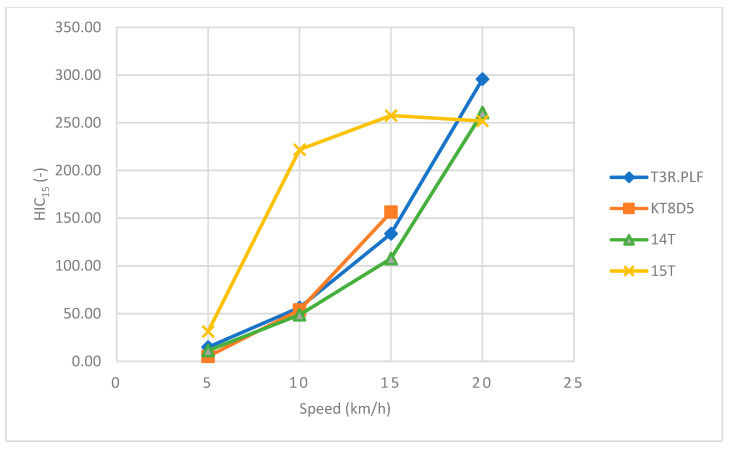
The influence of speed on the *HIC*_15_ for all four tram types during the primary impact.

**Figure 28 sensors-23-08819-f028:**
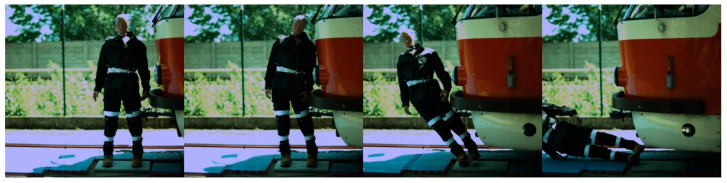
The side collision sequence between the ATD and T3R.PLF tram type approaching at a speed of 5 km/h.

**Figure 29 sensors-23-08819-f029:**
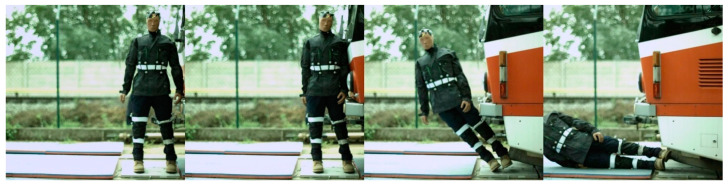
The side collision sequence between the ATD and KT8D5 tram type approaching at a speed of 5 km/h.

**Figure 30 sensors-23-08819-f030:**
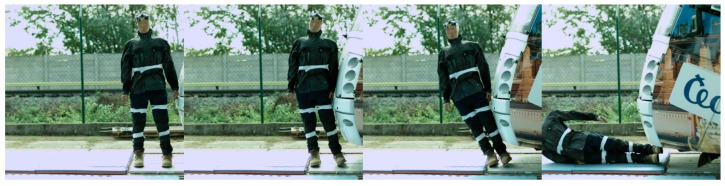
The side collision sequence between the ATD and 14T tram type approaching at a speed of 5 km/h.

**Figure 31 sensors-23-08819-f031:**
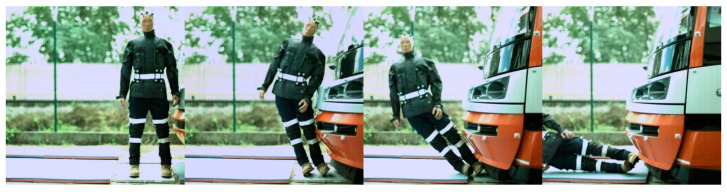
The side collision sequence between the ATD and 15T tram type approaching at a speed of 5 km/h.

**Figure 32 sensors-23-08819-f032:**
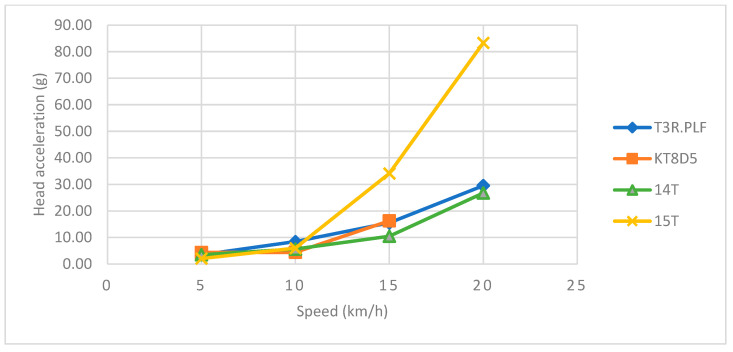
The influence of speed on the head acceleration for all four tram types during the primary impact in the side impact.

**Figure 33 sensors-23-08819-f033:**
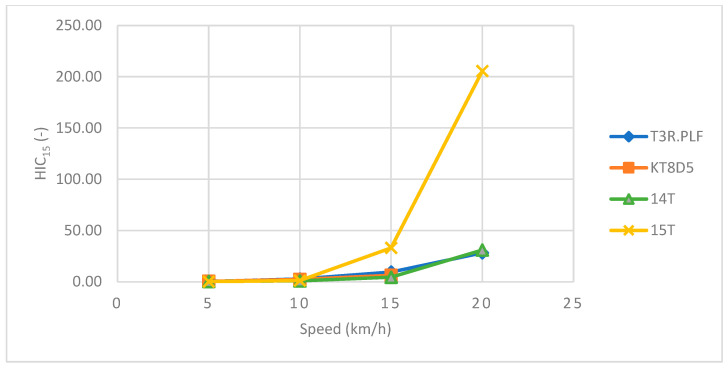
The influence of speed on the *HIC*_15_ for all four tram types during the primary impact in the side impact.

**Table 1 sensors-23-08819-t001:** Head injuries on the Abbreviated Injury Scale (AIS) [28,33].

HIC Range	AIS	Injury Level	A Detailed Description of a Head Injury
<135	0	No injury	No injury.
135–519	1	Minor	Abrasions and superficial lacerations on the skin and scalp; fracture of the nose; headache or dizziness; no loss of consciousness.
520–899	2	Moderate	Simple or decomposed fractures to the face; open fractures or displacements of the jaw; fractures of the jaw; unconscious for less than one hour.
900–1254	3	Serious	Different fractures; total loss of scalp; bruises to the cerebellum; unconscious for 1–6 h without severe neurological damages.
1255–1574	4	Severe	Complex facial fractures; exposure or loss of brain tissue; small epidural or subdural haematoma; unconscious for 6–24 h with severe neurological injuries.
1575–1859	5	Critical	Greater penetration of brain injuries; damage and haematoma to the trunk; epidural or subdural compression; axonal damage spread; unconscious for more than 24 h with critical neurological indications.
>1860	6	Fatal	Extensive destruction of both the skull and the brain; potentially fatal.

**Table 2 sensors-23-08819-t002:** The resulting head acceleration for frontal collision between the ATD and two older tram types for all four approaching speeds of the tram during the primary impact. The *HIC*_15_ values were converted into the probability of AIS3+ or AIS4+ head injuries.

	T3R.PLF				KT8D5			
Speed (km/h)	Max *a* (g)	*HIC* _15_	*p*(*AIS3+*) (−)	*p*(*AIS4+*) (−)	Max *a* (g)	*HIC* _15_	*p(AIS3+)* (−)	*p(AIS4+)* (−)
5	37.2	14.9	0.00	0.00	17.7	5.1	0.00	0.00
10	49.3	56.4	0.00	0.00	50.0	53.9	0.00	0.00
15	53.5	133.8	0.02	0.01	104.9	156.4	0.03	0.01
20	123.4	295.8	0.09	0.02	-	-	-	-

**Table 3 sensors-23-08819-t003:** The resulting head acceleration for frontal collision between the ATD and two modern tram types for all four approaching speeds of the tram during the primary impact.

	14T				15T			
Speed (km/h)	Max *a* (g)	*HIC* _15_	*p*(*AIS3+*) (−)	*p*(*AIS4+*) (−)	Max *a* (g)	*HIC* _15_	*p*(*AIS3+*) (−)	*p*(*AIS4+*) (−)
5	18.8	11.4	0.00	0.00	39.0	31.4	0.00	0.00
10	42.6	48.8	0.00	0.00	72.9	221.9	0.06	0.01
15	64.9	107.7	0.02	0.00	81.1	257.7	0.07	0.02
20	114.1	260.6	0.07	0.02	124.0	251.8	0.07	0.01

**Table 4 sensors-23-08819-t004:** The resulting head acceleration for side collision between the ATD and two older tram types for all four approaching speeds of the tram during the primary impact. The *HIC*_15_ values were converted into the probability of AIS3+ or AIS4+ head injuries.

	T3R.PLF				KT8D5			
Speed (km/h)	Max *a* (g)	*HIC* _15_	*p*(*AIS3+*) (−)	*p*(*AIS4+*) (−)	Max *a* (g)	*HIC* _15_	*p*(*AIS3+*) (−)	*p*(*AIS4+*) (−)
5	3.5	0.3	0.000	0.000	4.4	0.5	0.000	0.000
10	8.4	2.7	0.000	0.000	4.4	2.3	0.000	0.000
15	15.5	9.5	0.000	0.000	16.3	6.3	0.000	0.000
20	29.5	28.2	0.000	0.000	-	-	-	-

**Table 5 sensors-23-08819-t005:** The resulting head acceleration for side collision between the ATD and two modern tram types for all four approaching speeds of the tram during the primary impact.

	14T				15T			
Speed (km/h)	Max *a* (g)	*HIC* _15_	*p*(*AIS3+*) (−)	*p*(*AIS4+*) (−)	Max *a* (g)	*HIC* _15_	*p*(*AIS3+*) (−)	*p*(*AIS4+*) (−)
5	3.5	0.3	0.000	0.000	2.1	0.1	0.000	0.000
10	5.6	0.9	0.000	0.000	6.0	1.2	0.000	0.000
15	10.5	4.4	0.000	0.000	34.1	33.0	0.001	0.000
20	26.8	30.9	0.000	0.000	83.4	205.5	0.048	0.010

## Data Availability

The data presented in this study are available on request from the corresponding author with the permission of partners of the project. The data are not publicly available due to their technical meaning for partners of the project.

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
