# Peer review of "Pedestrian Safety in Frontal Tram Collision, Part 1: Historical Overview and Experimental-Data-Based Biomechanical Study of Head Clashing in Frontal and Side Impacts"

_sensors, 2023, doi:10.3390/s23218819_

Round 1
Reviewer 1 Report
Part 2 is more a test report than a research paper. The scientific purpose is unclear. It can be merged into Part 1.
The data drawing styles are not uniform in figures. The presentation of tables and figures is poor.
The present form cannot show enough novelty to publish in Sensors journal.
Some of sentences are difficult to read.
Author Response
Dear reviewer,
many thanks to you for all your helpful comments and recommendations. Please, find enclosed file with my reactions and revised article resubmitted to the website.
Kind regards
Frantisek Lopot

Reviewer 2 Report
The authors present a significant study on the experimental crash tests between the pedestrian ( represented by the Hybrid III ATD) and four types of trams. The research is very novel, this could be explained by the monetary price necessary to conduct such a study. However, there are some limitations ( especially with the utilization of the H3 dummy), which is mostly used as a passenger for the vehicle. There is a better dummy that could be used as a pedestrian ( Polar Pedestrian Dummy), nevertheless, it is not so easy to acquire this ATD. It could be useful to discuss the differences between H3 and the Polar dummy, by the authors. In Chapter 3 it should be explicitly stated that this study did not focus on secondary impact (perhaps cutting the Figure 9 after 400ms should be enough). The results are disseminated in an acceptable form and style, but it could be very valuable for future readers to place the files from the Qualisys system into some free online repository.
Author Response
Dear reviewer,
many thanks to you for all your helpful comments and recommendations. Please, find bellow enclosed file with my reactions and revised article resubmitted to the website.
Kind regards
Frantisek Lopot

Reviewer 3 Report
The authors have done a lot of work to conduct the research. The work is at a high level and, as the authors emphasize, these studies are quite innovative as no tram-pedestrian crash tests have been carried out so far. The results are satisfactory and the work is transparent. I have a few questions regarding the implementation of the research, the article does not show that the crash tests were repeated, so my question. Were comparative tests carried out in the same setting several times, if so, how much did the results differ? The second question was whether the difference in sampling data of ultra-fast cameras was not a problem with data analysis? Crash tests require equal frequency if we want to compare data. Were data from two different cameras compared in this case, or were they separate samples? Last question, were the dummy's joints calibrated before the crash tests? The manufacturer recommends calibrating after 5 crash tests.
Author Response

(The authors gave the same response as above.)

Round 2
Reviewer 1 Report
The novelty is still unclear, as dummy test is common, not as the author said it not carried out so far.
Not see revision.
Author Response
Dear reviewer,
we have further improved our article to satisfy your comments. However, I must also say that we did not receive very clear suggestions and instructions from your second review. Your second review is even more strict than the first one, despite all made improvements. The current improvements have therefore more cosmetic character. Regarding your comment about the use of dummies in crash tests - sure, it's a completely common procedure: in the automotive industry. We have no information that this is the case with trams or any rail vehicles. That's why we decided to design our publication as a trilogy, in order to present this topic, which is obviously becoming very acute at least in EU, in a slightly wider context. Regarding the level of English: our text has been re-proofread by a native speaker who is himself active in this research field.
I believe that the new version will already meet your expectations and conditions. If not, please provide a somewhat more specific comments and/or questions that we could respond to.
Thank you very much for the time you spend with our work.
I am looking forward to your reply.
With respect
Frantisek Lopot